# Role of Circadian Clock on the Pathogenesis and Lifestyle Management in Non-Alcoholic Fatty Liver Disease

**DOI:** 10.3390/nu14235053

**Published:** 2022-11-27

**Authors:** Nuria Perez-Diaz-del-Campo, Gabriele Castelnuovo, Gian Paolo Caviglia, Angelo Armandi, Chiara Rosso, Elisabetta Bugianesi

**Affiliations:** 1Department of Medical Sciences, University of Turin, 10126 Turin, Italy; 2Metabolic Liver Disease Research Program, I. Department of Medicine, University Medical Center of the Johannes Gutenberg-University, 55131 Mainz, Germany; 3Gastroenterology Unit, Città della Salute e della Scienza—Molinette Hospital, 10126 Turin, Italy

**Keywords:** NAFLD, clock disruption, sleep disturbances, chrononutrition

## Abstract

Several features of the modern lifestyle, such as weekly schedules or irregular daily eating patterns, have become major drivers of global health problems, including non-alcoholic fatty liver disease (NAFLD). Sleep is an essential component of human well-being, and it has been observed that when circadian rhythms are disrupted, or when sleep quality decreases, an individual’s overall health may worsen. In addition, the discrepancy between the circadian and social clock, due to weekly work/study schedules, is called social jetlag and has also been associated with adverse metabolic profiles. Current management of NAFLD is based on dietary intake and physical activity, with circadian preferences and other environmental factors also needing to be taken into account. In this regard, dietary approaches based on chrononutrition, such as intermittent fasting or time-restricted feeding, have proven to be useful in realigning lifestyle behaviors with circadian biological rhythms. However, more studies are needed to apply these dietary strategies in the treatment of these patients. In this review, we focus on the impact of circadian rhythms and the role of sleep patterns on the pathogenesis and development of NAFLD, as well as the consideration of chrononutrition for the precision nutrition management of patients with NAFLD.

## 1. Introduction

Non-alcoholic fatty liver disease (NAFLD) is a rising epidemic affecting around 32.4% of the global population [1]. The mechanism underlying the development and progression of NAFLD, as well as the prediction of hepatic and extrahepatic risk, is a complex area and is not yet fully understood [2]. Several risk factors, such as ethnicity, gender, age, body composition, weight, and the presence of metabolic syndrome can increase the risk of this liver disease [3]. In fact, the progression of chronic liver diseases, including NAFLD, occurs differently in men and women [4]. However, it seems that multiple etiopathogenic factors, such as obesity, genetics, environmental factors, insulin resistance, and changes in the gut microbiota, act in parallel or sequentially, causing NAFLD [5]. In this sense, following pioneer attempts, and in order to overcome the negatives attributed to NAFLD, it has been proposed to change the term NAFLD to MAFLD (metabolic dysfunction associated fatty liver disease) [6].

The epidemiological evidence in this field has suggested a close relationship between unhealthy lifestyle and NAFLD, making lifestyle correction the first-line approach in these patients [7]. In this regard, a link has been reported between the pathogenesis of chronic liver disease and some behaviors and activities, such as shift work, inadequate sleep, energy-dense fatty foods, or the exposure to artificial light [8]. The phenomenon of absence of synchrony between sleep and the circadian rhythm is called circadian misalignment and it has been associated with an increased incidence of metabolic and cardiovascular disorders [9]. Interestingly, sex differences in sleep appear in infancy and persist into childhood [4]. These underlying differences may be exacerbated at puberty and lead to objective sleep disturbances, which contribute to the higher prevalence of insomnia symptoms in young adolescents [10].

Circadian rhythms are controlled by the clock’s proteins and regulate the body’s functions [11], determining, for example, the most appropriate eating time. Concretely, the circadian clock contributes to the pathogenesis of NAFLD and non-alcoholic steatohepatitis because it enhances lipid dysregulation, oxidative stress, and inflammation [12,13]. Animal models have shown that the composition and timing of a diet influences the circadian clock and hepatic fat accumulation [14], leading to the awareness that for metabolic health it is not only important ‘what we eat’, but also ‘when we eat’ [15]. Since many genes controlled by the circadian clock are essential in the body’s metabolic processes, behavioral approaches, such as restricting the time of food intake, may have benefits in NAFLD independent of the weight loss effects [16]. Therefore, recent data recommend a lifestyle intervention to improve sleep duration and quality, along with diet and physical activity, for better metabolic benefits [17].

The purpose of this narrative review focuses on (i) the influence of circadian rhythms on the pathogenesis of NAFLD and (ii) on the role of sleep habits in the development of this liver disease. Finally, emerging evidence on chrononutrition and the use of fasting strategies to ‘synchronize’ lifestyle habits with the circadian clock in the management of patients with NAFLD will be reviewed.

## 2. Circadian Rhythms in Liver Metabolism

The rotation of the earth on its axis and around the sun determines regular changes in the environment with regards to light–dark cycles [18]. Many organisms have developed an endogenous “time-keeping” system in order to synchronize with external time signals (zeitgebers) [19]. The periodicity of this clock is approximately 24 h and is, hence, also called a circadian rhythm [20]. In mammals, the circadian pacemaker resides in the suprachiasmatic nuclei (SCN), and is controlled by specialized retinal ganglion cells that measure ambient light [8]. Clock proteins are the endogenous timing mechanisms that control these circadian rhythms and help to maintain circadian homeostasis [18]. Importantly, circadian clock proteins are also active in other organs (peripheral clocks), such as the lung or liver [21]. At the heart of the molecular complex are the core transcription factors Circadian Locomotor Output Cycles Kaput (*CLOCK*) and Brain and Muscle Arnt-like protein-1 (*BMAL1*) [22]. *CLOCK* and *BMAL1* drive the transcription of their own repressors, period (*PER*) and cryptochrome (*CRY*), resulting in a strongly self-regulated feedback loop [23] (Figure 1). During the daylight hours, increased transcription of *PER* and *CRY* genes results in the accumulation of the circadian repressors *PER* and *CRY* [24]. This process results in *CLOCK:BMAL1*-driven inhibition of *PER*, *CRY*, and clock-controlled genes. The highly controlled degradation of *PER* and *CRY* relieves transcriptional repression and allows *CLOCK:BMAL1*-mediated transcription to proceed again, establishing cycles in circadian gene expression [25]. 

The central clock in the SCN synchronizes with the peripheral clock to regulate liver metabolism [26]. The SCN is not sensitive to feeding patterns, whereas the liver peripheral clock depends on the feeding pattern for the amplitude and phase oscillation of its transcripts [27]. Concretely, hepatic metabolism processes, including glucose, lipid, and cholesterol/bile acid metabolism are highly dynamic, influenced by feeding/fasting and circadian rhythms [28]. For example, the release of the GLP-1, a peptide hormone produced in the intestinal epithelium which stimulates insulin and inhibits glucagon secretion, is circadian in both rodents and humans and this rhythmicity is lost with the consumption of a high fat/palmitate diet and with sleep deprivation [29,30]. Lastly, the production of hormones, such as melatonin or cortisol, also depends on the SCN rhythmic activity in response to light/darkness, whereas others hormones, such as FGF21, adiponectin, leptin, or insulin oscillate, following a circadian basis but are also regulated by feeding/fasting cycles [31]. In this regard, it has been demonstrated in a study of 110 individuals that the food intake timing relative to melatonin onset was significantly associated with the percentage of body fat (*p* = 0.02) and body mass index (*p* = 0.04), while controlling for sex [32]. 

Thus, in the absence of a robust hepatic circadian clock the body becomes more susceptible to metabolic disturbances, including insulin resistance or increased adiposity, which contribute to the pathogenesis of fatty liver diseases, diabetes, and obesity, as well as fibrosis and hepatocellular carcinoma [8,33]. In an animal model of male Wistar rats kept on a 12 h light/dark cycle, it was shown that forced activity during the sleep phase decoupled daily activity from the biological clock [34]. On the other hand, human studies have shown that dyssynchrony between feeding/breakfasting time and the circadian clock can lead to obesity and dysmetabolism [35]. Furthermore, it has recently been suggested that circadian rhythm alterations may have metabolic consequences in subjects carrying specific single nucleotide polymorphisms in other genes related to the circadian clock, such as *PNPLA3*, *PPARY*, *STAT3*, and *PPARGCα*, which, in turn, are related to NAFLD development and progression [18]. The circadian clock therefore plays an integral role in the coordination of metabolism, which has led to a closer examination of its contribution to a range of diseases, including NAFLD.

Additionally, it has also been described that disruption of circadian clock function can affect the gut microbiome [36,37]. Circadian rhythms have been shown to modulate the composition and function of commensal bacteria, as well as host susceptibility to bacterial and viral infections [38]. In this regard, a study of 14 male and female mice showed that disruption of the circadian clock by deletion of *BMAL1* abolished the rhythmicity of fecal microbiota composition in both sexes [39]. In the same study, *BMAL1* deletion also induced alterations in fecal bacterial abundance with different effects depending on sex. Moreover, some bacteria, such as cyanobacteria (*Synechococcus elongatus*), have their own circadian rhythms, providing a survival advantage to them [40]. Manipulating the circadian rhythms of gut microbial abundance and activity may therefore be a promising future approach.

## 3. Impact of Sleeping Disturbances on NAFLD Risk

### 3.1. Sleeping Habits and NAFLD

Sleep disturbance is an increasingly common attribute that has been implicated in the pathogenesis of chronic liver disease, particularly in the development and progression of NAFLD [41]. In fact, a systematic review and meta-analysis found a small, significant increase in the risk of NAFLD among subjects with less than 6 h of sleep duration [42]. Indeed, subjects with NAFLD have also been found to have poor sleep quality and delayed sleep onset [43], as also observed in cirrhotic patients [44]. In this context, a U-shaped distribution curve has been suggested for the relationship between sleep duration and NAFLD [45]. Nevertheless, the studies regarding the association between sleep characteristics and NAFLD have shown inconsistent and often contradictory outcomes, as demonstrated by a systematic review by Shen et al. where no association between either short or long sleep duration and fatty liver disease risk was found [46]. Concretely, in this study, short sleep has been defined as sleeping less than 5 h, less than 6 h or the lowest 25th percentile of sleeping hours, while long sleep duration was defined as sleeping more than 7 h and 8 h. 

Some plausible explanations for the relationship between sleep and metabolic diseases are related to alterations in the ability of ghrelin and leptin to signal the correct energy balance [47], or to the endocannabinoid system (ECS) which could be one of the mechanistic pathways through which sleep restriction promotes excessive food intake and obesity [17]. In this sense, the ECS has been suggested to be overactive in patients with obesity, promoting a state that favors metabolic processes leading to weight gain, lipogenesis, insulin resistance and dyslipidemia [48,49]. Thus, both disruption of the circadian cycle and sleep deprivation can affect energy balance and, over time, may bring about substantial changes in body composition [50]. Sleeping 4 h a night may lead to an increase in body weight compared to normal sleep [51], and so can an increase in appetite and calorie intake from unhealthy foods with a 20% higher calorie intake [52,53]. However, results from a meta-analysis of four humans’ RCT regarding the effect of sleep restriction on leptin and ghrelin showed no significant effects of sleep deficit on the two hormones (*p* = 0.84) [54]. 

In addition, other specific features of sleep pattern analysis, including the time required to fall asleep or daytime sleepiness, may have a relevant role in the development and progression of NAFLD [43]. A sleep duration of less than 5 h was shown to negatively influence the risk of developing NAFLD [55], which was also found in a study carried out in a group of 708 young, non-diabetic patients [56]. Furthermore, 4 h sleep restriction resulted in a reduction in insulin sensitivity and insulin response to glucose, indicating a weak β-cell response to increased insulin resistance [57,58] and implying an imbalance in cardiac sympathetic-vagal activity by altering 24 h cortisol levels, with higher evening peaks [17]. In contrast, a sleep duration between 7 and 9 h is inversely correlated with NAFLD presence (OR = 0.38, *p* = 0.05) [59]. In this sense, in a study by Tasali et al., 10 overweight adults, after being subjected to an extra 1.6 h of bedtime for 2 weeks, experienced reduced sleepiness, increased vigor and a decrease in overall appetite, particularly with regard to the desire for sweet and salty foods [60]. 

However, results also demonstrated the absence of an association between sleep and NAFLD and a protective role of short sleep duration (<6 h) on the onset of the disease, with a prevalence rate of NAFLD occurrence that increased significantly with increasing sleep duration in men (*p* = 0.02) [61]. Similarly, a prospective cohort study of 8965 subjects, of whom 2197 incident cases of NAFLD were identified, showed that subjects who slept 8–9 h (OR = 1.21) and ≥9 h of sleep per night (OR = 1.31) were significantly associated with elevated odds of NAFLD or increased risk of liver disease [62]. It is important to note that for the comparison of studies and outcomes we have to take into account the cohort and the type of study conducted, as well as the confounding factors included for outcome prediction. As a consequence, more studies in this field are needed to assess the relationship between circadian, behavioral, and conditioning factors, as well as the interaction of them, in patients with NAFLD.

### 3.2. The Role of Light and Social Jetlag on NAFLD

The SCN in the presence of light is able to modulate the liver and other cellular processes [63], creating an activity phase (daytime) and a resting phase (night-time). It has been shown that feeding leads to contrasting temporal signals of the SCN in rats during the resting phase [64]. In this context, shift work has been associated with a higher prevalence of NAFLD, especially when workers were continuously exposed, both in terms of duration and cumulative days, to night shifts [65]. Consistent with this, a study involving 758 workers (507 men, 251 women) with initially normal alanine amino transaminase (ALT) and a mean age of 32.9 years showed a 3.7-fold increase in higher levels of ALT in those workers who had persistent rotating shift work and baseline fatty liver ultrasound ALT [66]. Curiously, female workers with a pre-existing sonographic fatty liver had a significantly higher risk (OR = 8.5) of developing elevated ALT when exposed to p-RSW, compared to those without p-RSW exposure. Furthermore, a study by Wang et al. found that with increasing years of night shifts, the OR of ALT increased from 1.03 to 1.60 among workers without NAFL, suggesting that disruption of the circadian clock may exert a toxic effect on the liver [67]. Conversely, inconsistent results were found in a NHANES-based cross-sectional study, in which no association between shift work and increased risk of NAFLD was found [68]. 

Additionally, a major focus has been put on the exposition to artificial light at night (ALAN) [69]. Scientific evidence shows that ALAN can be a risk factor for body weight gain, suggesting an influence on the circadian clock [70] that, in turn, leads to pathological changes [71]. The data from the National Institute of Health on 43,000 women showed that women who slept with the light or television on were more likely to be obese at the beginning of the study, supporting the hypothesis that sleeping with the light or television on is associated with the development of obesity [72]. Thus, ALAN emerged as a statistically significant and positive predictor of obesity and being overweight [73]. Moreover, a study on mice showed that when kept in constant light exposure, they increase body mass and reduce glucose metabolism [74,75]. In addition, outdoor ALAN exposure has been suggested as a risk factor for breast cancer incidence (*p* < 0.01) [76], ALAN intensity and breast cancer rate (*p* < 0.05) [77] and ALAN and prostate cancer incidence (*p* = 0.0369) [78], while evidence is scarce for indoor light intensity at night [79]. On the other hand, in a systematic review and meta-analysis that assessed associations between ALAN exposure and breast cancer risk, it was shown that outdoor ALAN exposure was associated with a 12% increase in breast cancer risk, while indoor exposure was associated with a 13% increase, therefore producing similar effects. Importantly, outdoor ALAN is calculated from ground-level outdoor illumination measurements obtained from night-time satellite images which, however, do not provide information on light emissions [80], while for indoor ALAN all analyses were conducted with self-reported data, so there are no studies examining the effect using objective measurements [81]. Furthermore, it should be noted that the International Agency for Research on Cancer (IARC) has classified shift and/or night work in Group 2A of ‘probable carcinogens’ because they ‘involve circadian disorganization’ [82].

On the other hand, the imbalance between the circadian and social clocks can be quantified by the concept of social jetlag (SJL) [83]. SJL is a measure representing the imbalance between the circadian clock and the social clock, resulting from the time mismatch between sleep times on workdays and free days [84]. For example, a person used to going to bed and getting up early both on weekdays, due to social compromises, and on days off, suffers less from social jetlag [83]. It has been estimated that 50% of workers and students experience at least two hours of SJL, while as many as 70% experience at least one hour of SJL due to accumulated sleep deficit during the week or the presence of predetermined schedules forcing people to wake up earlier than they should/want to [85]. Social jetlag has also been found to significantly increase the likelihood of being overweight [84] and to have elevated blood pressure [86] and cardiovascular disease [87], which are directly associated with NAFLD. 

Indeed, a cross-sectional sample of 341 children showed that a 1h increase in SJL was associated with a 2.98% increase in body fat and a 0.90 kg/m^2^ increase in BMI [88]. Furthermore, in a longitudinal cohort of 1037 individuals in New Zealand, higher SJL was correlated with higher body mass index (β = 0.10, *p* = 0.012), more fat mass (β = 0.08, *p* = 0.031) and higher probability of being obese (OR = 1.2, *p* = 0.045) and meeting the criteria for a metabolic syndrome diagnosis (OR = 1.3, *p* = 0.031) [89]. A Dutch cohort study (*n* = 83) of 16-year-old adolescents showed a significantly higher BMI in subjects without SJL, compared to the group with more than 1 h of SJL (−0.81 kg/m^2^ vs. −2.09 kg/m^2^), and did not observe significant associations between social jetlag and obesity within one year [90]. Furthermore, in 710 young adults from 19 to 24 years old, individuals with SJL showed higher energy (*p* = 0.012), protein (*p* = 0.009), fat (*p* = 0.006), saturated fats (*p* = 0.014), MUFA (*p* = 0.022), and carbohydrate intake (*p* = 0.002) when compared to the individuals without SJL [91]. Curiously, in a cohort study of 145 apparently healthy participants, 2 h SJL was also correlated with higher resting heart rate and increased fasting cortisol compared to subjects with 1 h SJL [92]. Furthermore, 2 h of SJL were also associated with an approximately 2-fold increased risk of pre-diabetes and type-2 diabetes mellitus in 1585 subjects from the New Hoorn Study cohort [93]. Furthermore, a retrospective, longitudinal study of 625 individuals confirmed these results, showing that SJL was positively associated with fasting glucose (β = 0.30, *p* = 0.0001) and triglyceride (β = 0.22, *p* = 0.01) levels in the type-2 diabetes mellitus group [94]. Indeed, a study carried out in 447 participants, who worked part- or full-time day shifts, concluded that SJL was associated with a lower high-density lipoprotein–cholesterol level, higher triglycerides, fasting plasma insulin, insulin resistance and adiposity (*p* < 0.05), after adjustment for covariates [95]. Finally, associations between SJL and depressive symptoms have also been reported in an essentially rural population (*n* = 4051), especially in 31 to 40 year olds [96]. 

Consequently, a decrease in SJL could be beneficial in terms of improved health and performance, especially for those who experience a considerable discrepancy (more than 2 h) between their circadian and social clocks. All these metabolic changes may influence the risk of developing NAFLD and should, therefore, be taken into account in the personalized management of these patients.

## 4. Chrononutrition in the Management of NAFLD

Increasing evidence suggests that our circadian clock interacts with metabolic functions and that meal timing is an important factor in metabolic regulation [32,97]. Chrononutrition is a new discipline that investigates the relationship between circadian rhythms, nutrition, and metabolism, and has developed rapidly in recent years [98]. It has clearly been demonstrated that when we eat is as important as what and how much we eat for the progression of chronic diseases [99]. The eating behavior of our modern society is often characterized by prolonged and irregular daily eating patterns which, together with the Western-style diet, sedentary lifestyle, and chronic sleep deprivation, may contribute to an increased risk of metabolic diseases, including NAFLD [100,101]. Concretely, late-night eating (defined as eating dinner within 2 h of bedtime) is associated with increased risk of metabolic diseases [102].

The chronotype (with chronos meaning time in Greek) is defined as an individual’s circadian preference for activity being determined by the expression of at least a dozen core circadian genes [103]. Individuals’ chronotype can be categorized as “early” or “morning”, when people go to bed and wake up early, whereas “late” or “evening” will define people who go to bed and wake up late [104,105]. A recent study has shown that individuals with a late chronotype presented a significantly higher visceral adiposity index, liver fat equation, hepatic steatosis, and non-alcoholic steatohepatitis indexes [106], and a lower adherence to a healthy diet, calculated with the Chrononutrition Profile Questionnaire [107]. Moreover, people with late chronotypes reported 0.25 less servings of fruit and 0.13 less servings of vegetables per day compared to people with early chronotypes [108]. Similarly, in 194 patients with type-2 diabetes mellitus who do not work shift work, a late chronotype was associated with skipping breakfast (*p* = 0.014) [109]. In fact, it has been demonstrated that frequent breakfast skipping was independently associated with higher cardiovascular mortality risk (*p* < 0.001) in 3004 participants with metabolic associated fatty liver disease (MAFLD), but not in MAFLD-free individuals (*p* = 0.280) [110]. 

Current management of NAFLD includes diet and lifestyle changes for achieving weight loss [111]. However, the difficulty of adhering to diets, as well as the current “obesogenic” environment, characterized by physical, social, and cultural factors that encourage sedentary lifestyles and obesity, lead to the alteration of the biological clock [112]. In this regard, some studies have suggested alternative approaches to reducing the risks associated with metabolic diseases, such as intermittent fasting (IF) or time-restricted feeding (TRF) [113,114], which have been recently reviewed in detail in NAFLD [115]. In fact, it has been demonstrated that these nutritional strategies are useful for realigning lifestyle behaviors with circadian biological rhythms [116] (Figure 2). 

Firstly, it should be noted that TRF is not synonymous with IF. Intermittent fasting usually refers to a period of water-only or very low-calorie fasting lasting of less than 24 h, followed by a period of normal eating for one or two days to induce changes in metabolism [15]. However, it may also include fasting on alternate days, eating little or no food for two consecutive days (5:2) or periodic fasting (2–21 days of minimal calories), followed by normal eating on non-restricted days [15,113]. Caloric intake in this type of diet is strictly regulated on fasting days. In this regard, an umbrella review of 11 meta-analyses of 130 randomized clinical trials has described beneficial associations of IF with anthropometric and cardiometabolic outcomes, supporting the use of this approach in overweight or obese adults [114]. Importantly, both IF and calorie-restricted diets have been shown to be equally effective in managing weight loss and improving cardiometabolic outcomes [117]. However, some benefits of IF, such as promoting ketogenesis, being linked to circadian biology and showing pronounced weight loss outcomes in individuals with elevated body mass index, have made it emerge as a promising strategy in patients with cardiovascular diseases [118]. In addition, a meta-analysis to estimate the effects of IF in adults with NAFLD confirmed an improvement in ALT (*p* ≤ 0.00) and AST (*p* ≤ 0.00) compared to a non-fasting group [101]. These results appear to be related to a decrease in body weight and a reduction in body mass index. In fact, previous meta-analyses have shown that subjects following IF had greater reductions in both body weight and body mass index compared to the control group [119,120]. 

In contrast, TRF is a dietary approach defined as feeding within a period of ≤10 h and fasting for at least 14 h a day, where feeding is consolidated to the active period, aligning peripheral and central circadian rhythms [121]. This strategy could be divided into early (8 a.m. to 4 p.m.) and late (1 p.m. to 9 p.m.) [122]. When applied to human populations, TRF does not restrict the number of calories consumed or require a dietary profile of macronutrients, making it easier for patients to adopt. Indeed, the use of this strategy has been reported to improve weight loss, cardiometabolic health, and overall wellbeing [123]. A study in mice showed that TRF is able to alleviate metabolic diseases with benefits proportional to the duration of fasting, due to its pleiotropic beneficial effects on metabolism, such as reducing insulin signaling and switching energy utilization from glucose to fat during fasting, or reducing biomarkers of inflammation [124]. In humans, a 12-week RCT demonstrated that fasting for 16 h (16/8 TRF), combined with a low-sugar diet, was associated with a significant reduction (*p* < 0. 05) in body fat and body weight, as well as circulating levels of fasting blood glucose and liver biomarkers (ALT: 34 ± 13.9 to 21.2 ± 5.4 U/L; AST: 26.3 ± 6.2 to 20.50 ± 4 U/L; γ-glutamyl transpeptidase: 33 ± 15 to 23.2 ± 11.1 U/L) compared to the control group [125]. In this study, a significant decrease (*p* < 0.05) was also observed in the fibrosis score (6.3 ± 1 to 5.2 ± 1.2 kPa) and in the controlled attenuation parameter (322.9 ± 34.9 to 270.9 ± 36.2 dB/m). However, to date only one study conducted on 15 men compared the effects of early TRF and late TRF, finding only a difference in mean fasting blood glucose, which was lower in early TRF (*p* = 0.02) [126].

In summary, both IF and TRF could be easily adoptable behavioral interventions to prevent/act on outcomes in patients with metabolic syndrome and NAFLD [127]. However, it is important to keep in mind some limitations of these nutritional strategies, such as the difficulty of maintaining the daily feeding window of 10–6 h while maintaining family, social, or work life [15]. Another major limitation is the lack of standardization of protocols for both diets and the scarce research in NAFLD patients, so no clear conclusions can be established at present.

## 5. Conclusions

Sleep disturbances have been implicated in the pathogenesis of NAFLD, influencing patients’ health and quality of life. The misalignment with the circadian rhythm thus adds to the metabolic burden of this complex disease. Understanding the molecular mechanisms involved in the control of the circadian clock is therefore essential not only to advance scientific knowledge, but also to improve public health by identifying new therapeutic targets and lifestyle modifications.

Indeed, public strategies aimed at optimizing shift work schedules and respecting biological clocks may be useful to reduce social jetlag. Along with these strategies, the message of the importance of taking care of the details of sleep should be reinforced, starting with reducing artificial lighting and ending with observing social schedules respecting the hormonal rhythm. Therefore, integrative analyses of precision variables should include age, sex, body phenotype, and lifestyle variables, together with personalized issues (genetics, epigenetics, microbiota composition, etc.), in order to contribute to the prescription of more personalized treatments. However, as research in this field is limited, future studies should be conducted to propose safe and effective feeding strategies to improve the management of these patients.

## Figures and Tables

**Figure 1 nutrients-14-05053-f001:**
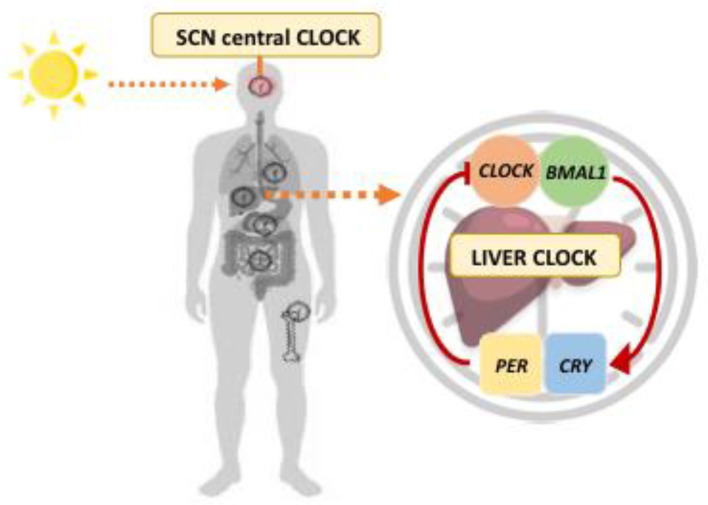
Transcriptional feedback loop constituting molecular biological clocks. The central clock of the SCN, driven by light, synchronizes the clocks of peripheral tissues, including the liver. At the heart of the molecular complex are the clock proteins *CLOCK* and *BMAL1* that heterodimerize to induce transcription of the *PER* and *CRY* genes. This feedback loop takes approximately 24 h to complete and is the molecular basis of the mammalian biological clock. *BMAL1*: Brain and Muscle Arnt-like protein-1; *CLOCK*: Circadian Locomotor Output Cycles Kaput; *CRY*: Cryptochrome; *PER*: Period; SCN: Suprachiasmatic Nucleus.

**Figure 2 nutrients-14-05053-f002:**
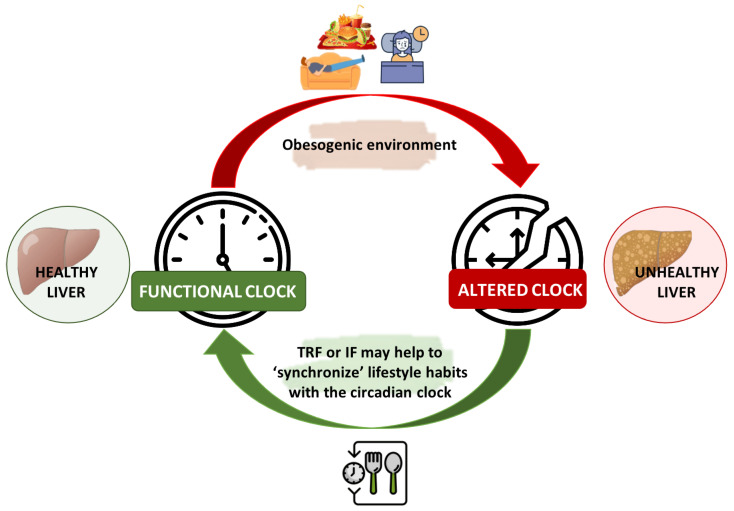
Chrononutrition in the alignment of the liver’s circadian rhythm. Current “obesogenic” environments lead to the disruption of the biological clock. In this context, meal timing and dietary components play an important role in the regulation of circadian clocks. Nutritional strategies, such as intermittent fasting (IF) or time-restricted feeding (TRE) have been demonstrated to be useful in realigning lifestyle behaviors with circadian biological rhythms, thereby enhancing metabolic health and reducing the risk of NAFLD. IF: Intermittent Fasting; TRF: Time-Restricted Feeding.

## Data Availability

All data are publicly available.

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
