# Peer review of "Role of Circadian Clock on the Pathogenesis and Lifestyle Management in Non-Alcoholic Fatty Liver Disease"

_nutrients, 2022, doi:10.3390/nu14235053_

Round 1

Reviewer 1 Report

Thanks for the opportunity to review this interesting manuscript titled “Role of the Circadian Clock on the pathogenesis and Lifestyle Management in /non-Alcoholic Fatty Liver Disease”. The concepts of circadian rhythm and circadian rhythm misalignment in metabolic syndrome and NAFLD are undoubtedly important areas for further study. Please review the following comments as I hope including further details will increase the nuance of the manuscript.

Could the authors add more details to describe the mechanism of the following statement if they are known: “or to the endocannabinoid system, which could be one of the 169 mechanistic pathways through which sleep restriction promotes excessive food intake 170 and obesity [14]”

Please discuss this further: “Sleeping 4 hours a night may lead to an increase in body weight 171 compared to normal sleep [45],”. Is it possible that behavioural factors and conditioning may lead to over-eating rather than circadian driven factors?

This sentence does not flow, please edit: “In this sense, a study by Tasali et 185 al. 10 overweight adults underwent an extra 1.6 hours of bedtime for 2 weeks, resulting in less sleepiness, more vigorous  (more vigourpus what?)and a decrease in overall appetite, particularly in the desire for sweet and salty foods [54]”

Please present some thoughts about the discordant relationships identified between sleep patterns and NAFLD development.

Regarding the following statement, and subsequent discussion, please describe whether the increased liver biochemistry markers were associated independent of weight gain? Describe the study design, and cite the limitations if weight gain was not discussed. “In line with this, a study 204 evidenced a relationship between shift work and higher levels of aspartate 205 aminotransferase (AST) and alanine transaminase (ALT)”.

Please discuss why indoor ALAN may not be associated with similar risks as outdoor ALAN?

Please reference the following manuscript , and cite that IF and time restricted eating have been recently reviewed in detail in NAFLD.

PMID: 36364915

·       DOI: 10.3390/nu14214655

Author Response

We thank the reviewer for his/her valuable comments which help us to improve our manuscript. Enclosed the response to the comments

Reviewer 2 Report

The manuscript by Perez-Diaz-del-Campo is a very well-written review on a very timely topic such as the role of circadian clock on the pathogenesis and lifestyle management in NAFLD. It would be interesting to complete this review with some missing information.

One interesting point to be considered is about the gender/sex influence on the impact of sleep, the role of light and social jetlag and strategies of chromonutrition in the development of NAFLD. Are there any studies addressing this point? If not, that should be disclosed in this review. 

In the conclusions the authors state that patients with NAFLD may present sleep disturbances. However, throughout the manuscript the authors suggest sleep disturbances are implicated in the pathogenesis of NAFLD. Can the authors discuss about the casualty between NAFLD and sleep disturbances or the other way around.

Minor comments:

It is recommended that the authors add the term MAFLD, a consensus-driven proposed nomenclature for NAFLD.

Regarding the incidence of NAFLD it has been reported to be somehow higher that 25% in the last years (PMID: 35798021).

Author Response

We thank the reviewer for his/her valuable comments which help us to improve our manuscript. Enclosed the response to the comments.

Round 2

Reviewer 1 Report

The authors have addressed all of my concerns, and I believe it is a novel contribution to the field.